# Multi-Omics Analysis of *Curculio dieckmanni* (Coleoptera: Curculionidae) Larvae Reveals Host Responses to *Steinernema carpocapsae* Infection

**DOI:** 10.3390/insects16050503

**Published:** 2025-05-07

**Authors:** Bin Wang, Fanyu Meng, Shiqi Song, Bin Xie, Shuxia Jia, Dongying Xiu, Xingpeng Li

**Affiliations:** 1Jilin Provincial Key Laboratory of Insect Biodiversity and Ecosystem Function of Changbai Mountains, Beihua University, Jilin 132013, China; wb2651747251@outlook.com (B.W.); m1040910@outlook.com (F.M.); songshiqi0107@outlook.com (S.S.); x1055957222@outlook.com (B.X.); jsxlxp@hotmail.com (S.J.); 2Jilin Academy of Forestry Sciences, Jilin 132000, China; xdyzyn@hotmail.com

**Keywords:** *Steinernema carpocapsae*, *Curculio dieckmanni*, multi-omics analysis, insect-EPN interactions

## Abstract

Hazelnut weevil larvae (*Curculio dieckmanni*) cause substantial damage to hazelnut crops, and their concealed feeding behavior makes them difficult targets for chemical insecticides. The entomopathogenic nematode *Steinernema carpocapsae*, known for its ability to effectively kill weevil larvae, presents a promising alternative for biological control. To investigate the molecular basis of larval responses to EPN infection, we analyzed gene and protein expression changes in hazelnut weevil larvae following exposure to *S. carpocapsae*. Our results revealed significant alterations in immune-related genes and metabolic pathways, highlighting how larvae activate defensive mechanisms and strategically reallocate energy resources during infection. These insights enhance our understanding of insect resistance mechanisms and could facilitate the optimization of EPN-based biological control strategies for hazelnut weevil management.

## 1. Introduction

*Steinernema carpocapsae* is an entomopathogenic nematode (EPN) widely used in biological control [1]. It exhibits a typical “ambush” foraging behavior and can effectively parasitize different soil-dwelling insect pests [2]. Recent studies have shown that *S. carpocapsae* can rapidly kill its host by releasing its bacterial symbiont, *Xenorhabdus nematophila* [3]. EPNs of the genus *Steinernema* invade the host, and the rapidly released *X*. *nematophila* from EPNs of the genus *Steinernema,* after invading the host, evade recognition by host immune hemocytes, inhibit cellular phagocytosis and nodule formation by evading recognition of host immune hemocytes through surface components such as flagella and pilin [4]. Subsequently, the nematodes secrete proteolytic substances to inhibit phenoloxidase activity or release venom proteins containing a variety of enzymes and toxins to modulate the host’s immune response [5,6]. Meanwhile, the symbiotic bacteria inhibit the cellular immune response of the host by interfering with the arachidonic acid signaling pathway by secreting 31 kDa protein toxin and lipopolysaccharides (LPS), which lead to the degradation of tissue structures and rapid pathogen reproduction and, ultimately, host mortality [7,8,9].

In response to EPN *S. carpocapsae* infection, insects activate various physiological and molecular defense mechanisms to counteract pathogen-induced damage [8]. The host insect initiates immune recognition mechanisms when the infective juveniles (IJs) of *S. carpocapsae* successfully enter the hemocoel. Pattern recognition receptors (PRRs) activate a series of cellular and humoral immune responses when they recognize pathogen-associated molecular patterns (PAMPs) [8]. Cellular responses are mediated by hemocytes limiting pathogen spread and replication through executing phagocytosis, nodulation, and encapsulation of invading nematodes [10,11]. However, *S. carpocapsae* and symbiotic bacteria effectively avoid recognition and phagocytosis by the host’s hemocytes through cuticle camouflage and the secretion of immune-suppressive molecules [8,12]. Transcriptomic analysis revealed that the expression of genes related to cellular immunity in *Drosophila suzukii* infected with *S. carpocapsae* was significantly downregulated, including genes in the Jak/STAT signaling pathway associated with encapsulation and hemocyte differentiation [3]. At the level of humoral immunity, the prophenoloxidase (proPO) system is an important pathway for insects to cope with pathogens, and it can encapsulate and eliminate invading pathogens through the melanization reaction [8,13]. For instance, RNA sequencing analysis showed that the genes PPO1, PPO2, and pro-PO A1 in larvae of *Drosophila* were upregulated when infected both by symbiotic and axenic *S. carpocapsae*, while the genes for black cells and phenoloxidase subunit A3 were only upregulated in response to infection by axenic nematodes [14]. Function loss of these genes (PPO1, PPO2, and PPO3) leads to reduced survival of larvae when infected both by axenic and symbiotic *S. carpocapsae* [15]. In addition, phenoloxidase (PO) activity in larvae of *Drosophila* increased greater when infected by axenic *S. carpocapsae* than in response to infection by symbiotic *S. carpocapsae*. Similarly, the expression levels of serine protease P56, prophenoloxidase activation factor 1, PPO, and serine protease inhibitor 28 in *Octodonta nipae* infected with *S. carpocapsae* were generally downregulated at all times [16].

Furthermore, *S. carpocapsae* infection can also induce the expression of genes encoding antimicrobial peptides (AMPs) and transcription expression of genes regulated by Toll or Imd signaling in insects [17]. For instance, the serine protease inhibitor gene Serpin-27A is upregulated by symbiotic nematodes and downregulated by axenic nematodes, while the *Drosophila* IKKγ kenny and transcription factor Relish are upregulated by axenic nematodes, while the *Drosophila* IKKγ kenny and transcription factor Relish are upregulated by both type of nematodes [14]. Notably, the transcriptional activation of AMP genes such as Attacin, Diptericin, Drosomycin, and Metchnikowin in the host only occurs during infection by symbiotic nematodes [17]. The symbiotic bacteria *X. nematophila* can reduce the effectiveness of these antimicrobial peptides by secreting specific toxins, thereby suppressing the host’s antimicrobial capacity [8].

The interaction between EPNs and insect hosts involves the regulation of multiple immune signaling pathways, including the Imd, Toll, JNK, JAK/STAT, and TGF-β pathways [18]. Particularly, the host’s Imd pathway is significantly activated to initiate a specific antibacterial response during the early stages of pathogen infection and the phase after bacteria release, but the overall immune response remains limited [13]. In addition, the gene expression profile of insects undergoes significant changes when infected by *S. carpocapsae* and *X. nematophila*. For example, the transcriptomes of the fat body, hemocytes, and midgut in *Spodoptera frugiperda* show differential expression following infection, reflecting the specific response of the host to the pathogen in different tissues. These changes may involve the upregulation or downregulation of immune-related genes, as well as pathways associated with metabolism and stress [19]. EPN infection also significantly disrupts the normal metabolic functions of the host, such as the expression of heat shock proteins (HSPs), detoxifying enzymes, and peroxidase genes. These changes indicate that the host is in a state of stress, which indirectly weakens the insect’s defensive capabilities [3]. In summary, the resistance response of insects to *S. carpocapsae* and its symbiotic bacteria *X. nematophila* involves the participation of physical barriers, cellular and humoral immune mechanisms, gene expression regulation, and multiple immune signaling pathways. The EPNs achieve successful infection by suppressing the defense mechanisms of the host at multiple levels. A deep understanding of these mechanisms will help optimize the application of entomopathogenic nematodes in integrated pest management.

The hazelnut weevil, *Curculio dieckmanni* (*Curculionidae*), is a destructive pest of hazelnut trees in China [20]. It is difficult to control due to the larvae’s concealed feeding behavior within nut kernels [21]. Moreover, pesticide application results in residues in nuts and significant environmental contamination [22]. Our previous research demonstrates that *S. carpocapsae* is highly virulent to *C. dieckmanni* larvae, killing most larvae within 24 h. Therefore, *S. carpocapsae* presents a promising biocontrol agent against *C. dieckmanni*.

In this study, we employed integrated proteomic and transcriptomic analyses to investigate the molecular responses of *C. dieckmanni* larvae to *S. carpocapsae* infection. The objective was to characterize the gene expression changes at both mRNA and protein levels, elucidate functional genes and regulatory mechanisms, and uncover the molecular pathways associated with insect resistance to EPN infection. This study enhances our understanding of host–pathogen interactions and contributes to optimizing EPN-based biocontrol strategies.

## 2. Materials and Methods

### 2.1. Study Cohort

In September 2024, hazelnuts infested by *Curculio dieckmanni* larvae were collected from a hazelnut orchard in Gangyao town, Jilin Province, China. The damaged hazelnuts infected by larvae were evenly placed on sterilized soil (30 cm deep) in plastic pots (upper diameter: 60 cm, height: 40 cm), which featured evenly distributed circular ventilation holes (0.3 cm in diameter) on the bottom and sides. After 1 to 3 days, most weevil larvae emerged from the nuts and burrowed into the soil. The soil surface was periodically sprayed with sterile water to maintain humidity levels similar to field conditions. The pots were kept under natural temperature and light conditions.

The *Steinernema carpocapsae* All nematode population was obtained from Hongrun Agricultural Technology Co., Ltd. (Weifang, China). Nematodes were propagated in fifth-instar *Galleria mellonella* larvae and collected using modified White traps [23]. After being rinsed three times with sterile water, the nematodes were stored in 100 mL of autoclaved distilled water (the concentration of nematodes (IJs) in storage was about 1000/mL) at 14 °C in vented culture flasks until use [24].

### 2.2. Experimental Design

Weevil larvae were separated from the soil using a 40-mesh sieve. The larvae were rinsed with sterile water, disinfected with 75% alcohol, blotted dry with filter paper, and then weighed. A nematode suspension was prepared by diluting *S. carpocapsae* in PBS buffer to achieve a concentration of 2 nematodes per milligram of weevil body weight. A total of 10 μL of the infective juvenile nematode suspension was injected between the second and third abdominal segments of each larva using a microliter syringe. The experiment comprised two groups: one group of larvae was injected with the nematode suspension, while the control group was treated with an equivalent volume of PBS buffer (the same solvent as the nematode suspension). There were three biological replicates per group, and each replicate contained 15 to 20 weevil larvae, with a total of 132 larvae included in the study. Twelve hours after treatment, all larvae were individually transferred to 1.5 mL centrifuge tubes and stored at −80 °C for subsequent analyses. The insect body powder for each replicate ground in liquid nitrogen was collected for omics analysis.

### 2.3. Transcriptomic Analysis

Total RNA was extracted from homogenized larvae powder (about 100 mg per biological replicate) using a TRIzol reagent (Tiangen Biotech, Beijing, China). The mRNA fraction was further purified using a column-based insect mRNA extraction kit (Tiangen Biotech, Beijing, China) following the manufacturer’s protocol, which included an optional DNase digestion step. The mRNA concentration and purity were assessed with a Nanodrop 2000 spectrophotometer (Thermo Fisher Scientific, Wilmington, DE, USA), and mRNA integrity was evaluated using an Agilent 2100 Bioanalyzer (Agilent Technologies, Santa Clara, CA, USA) with an mRNA 6000 Nano LabChip kit (Agilent Technologies, Santa Clara, CA, USA). Samples with an RNA Integrity Number (RIN) greater than 7.0 were used for subsequent library preparation. mRNA-seq libraries were prepared using the BGI mRNA Library Prep Kit (DNBSEQ) (Beijing Genomics Institute, Shenzhen, China), with the incorporation of unique barcode sequences for each sample identification. Library quality and quantity were validated using the Agilent 2100 Bioanalyzer with a DNA 1000 LabChip kit (Agilent Technologies, Santa Clara, CA, USA) and quantified via quantitative PCR (qPCR). Pooled libraries were sequenced on the DNBSEQ-T7 platform (BGI, Shenzhen, China) using DNA Nanoball (DNB) technology, generating 100 bp single-end reads. Raw sequencing reads were subjected to quality control using FastQC software (v. 0.12.0). Low-quality reads and adapter sequences were trimmed with Trimmomatic (v. 0.40). The cleaned reads were mapped to the reference genome using HISAT2 software (v. 2.2.1), and transcript assembly was performed using StringTie (v. 2.2.3). Differential gene expression analysis was conducted using DESeq2 software (v. 1.49.0) with the ead counts normalized via the median-of-ratios method.

Differentially expressed genes (DEGs) were identified based on the thresholds of an adjusted *p*-value (false discovery rate, FDR) ≤ 0.05 and |log_2_ fold change| ≥ 1. Raw RNA-seq data have been submitted to the NCBI Sequence Read Archive (SRA) database under accession number PRJNA1199052. Differential gene expression was analyzed as described by Love et al. (2014) [25].

### 2.4. Proteomic Analysis

#### 2.4.1. Protein Extraction and TMT Labeling

Larvae used for mRNA-seq were also subjected to proteomic analysis (about 300 mg per biological replicate). Proteins were extracted using phenol extraction buffer (containing 10 mM DTT and 1% protease inhibitor) followed by ultrasonication. Equal volumes of Tris-saturated phenol were added and then centrifuged at 5500× *g* for 10 min at 4 °C. The upper phenol phase containing proteins was collected and precipitated overnight at −20 °C by adding five volumes of 0.1 M ammonium acetate in methanol. Protein pellets were washed sequentially with cold methanol and acetone and redissolved in 8 M urea buffer. The protein concentration was determined using a BCA assay kit (Beyotime Biotechnology, Shanghai, China). Protein integrity was confirmed via SDS-PAGE.

Extracted proteins (100 µg/sample) were subjected to trypsin digestion, as described previously by Batalha et al. (2012) [26]. The resulting peptides were desalted using Strata X C18 columns (Phenomenex, Torrance, CA, USA), lyophilized, and then dissolved in 0.5 M triethylammonium bicarbonate (TEAB). Peptide labeling was performed using the TMT labeling reagent (Thermo Fisher Scientific, Waltham, MA, USA) according to the manufacturer’s instructions. Following incubation at room temperature for 2 h, the labeled peptides were again desalted and lyophilized.

#### 2.4.2. High-pH Reverse-Phase Fractionation (HPLC)

The labeled peptides were fractionated using high-pH reverse-phase chromatography (Agilent 1260 Infinity II HPLC) (Agilent Technologies, Santa Clara, CA, USA) with an Agilent 300Extend C18 column (250 mm × 4.6 mm, 5 µm particle size). The peptides were eluted at pH 9 with an acetonitrile gradient (8–32%) over 60 min and combined into 14 fractions, then lyophilized prior to mass spectrometry.

#### 2.4.3. LC-MS/MS Analysis

The peptides were analyzed using an EASY-nLC 1200 UPLC system (Thermo Fisher Scientific, Bremen, Germany) coupled to a Q Exactive™ Plus mass spectrometer (Thermo Fisher Scientific, Waltham, MA, USA). The lyophilized peptides were dissolved in solvent A (0.1% formic acid, 2% acetonitrile), separated at a flow rate of 450 nL/min with a linear gradient: 8–22% solvent B (0.1% formic acid in 90% acetonitrile) over 0–20 min; 22–35% B from 20–33 min; 35–80% B from 33–37 min, and holding at 80% B until 40 min.

The peptides were ionized via nano-electrospray ionization (NSI) at 2.2 kV and analyzed in the data-dependent acquisition (DDA) mode. Full-scan mass spectra were acquired over the range 400–1500 *m*/*z* at a resolution of 70,000. The top 20 precursor ions per scan were selected for higher-energy collisional dissociation (HCD) fragmentation, with a collision energy of 30%, resolution of 17,500, an automatic gain control (AGC) target of 5 × 10^4^, an intensity threshold of 6.3 × 10^4^ ions/s, a maximum injection time of 50 ms, and dynamic exclusion set at 30 s.

#### 2.4.4. Protein Identification and Quantification

The acquired mass spectrometry data were analyzed using MaxQuant software (v1.6.15.0). Searches were conducted against a custom database derived from the transcriptome data generated in this study. Enzyme specificity was set as trypsin/P with a maximum of one missed cleavage, and the minimum peptide length was seven amino acids. Carbamidomethylation (CAM) on cysteine residues was defined as a fixed modification, and oxidation (M) and acetylation (protein N-term) as variable modifications. Mass tolerance for precursor ions was set at 20 ppm for the first search and 5 ppm for the main search, with the fragment ion tolerance at 0.02 Da. The false discovery rate (FDR) threshold for peptide and protein identification was set to 1%, and protein identification required at least one unique peptide.

#### 2.4.5. Bioinformatics and Statistical Analysis

Proteins identified were functionally annotated using NR (NCBI non-redundant protein database), Swiss-Prot, GO (Gene Ontology), KEGG (Kyoto Encyclopedia of Genes and Genomes), Pfam, and eggNOG databases. Differentially expressed proteins (DEPs) were screened based on fold-change thresholds (>1.3 or <1) and significance levels (*p* < 0.05). GO and KEGG enrichment analyses were performed using eggnog-mapper (v2.0.0) and DIAMOND software (v2.1.6). Statistical significance for enrichment analysis was determined using Fisher’s exact test (*p* < 0.05).

### 2.5. Integrated Transcriptomic and Proteomic Analysis

To explore the correlation between transcriptomic and proteomic data, protein IDs were mapped to corresponding transcript IDs, and statistical analyses were conducted to evaluate the overlap between the two datasets. Expression levels of the transcripts and proteins were integrated and converted to Log_2_ fold-change (Log_2_FC) values. Differential expression was defined using the following criteria: for transcripts, a Log_2_FC > 1 and *p*-value < 0.05 indicated upregulation, while a Log_2_FC < −1 and *p*-value < 0.05 indicated downregulation; for proteins, a fold change > 1.5 (*p*-value < 0.05) indicated upregulation, whereas a fold change < 1/1.5 (*p*-value < 0.05) indicated downregulation.

### 2.6. Real-Time Quantitative PCR

To validate the integrated transcriptomic and proteomic results, selected genes were analyzed using real-time quantitative PCR (qRT-PCR). The same mRNA samples used for mRNA-seq were employed for qRT-PCR. First-strand cDNA was synthesized using the TransScript^®^ All-in-One First-Strand cDNA Synthesis SuperMix (Kemi Biotech Co., Ltd., Changchun, China). Each sample was analyzed in triplicate, with β-actin and GAPDH serving as internal reference genes. A melting curve analysis was performed, and relative gene expression levels were calculated using the 2^−ΔΔCT^ method [27]. To make the data normally distributed, log-transformed of 2^−ΔΔCT^ (log(2^−∆∆Ct^)) was used in our result. The primer sequences, qRT-PCR reaction details, and amplification conditions are provided in Appendix A.

## 3. Results

### 3.1. Sources of Variation in mRNA and Protein Expression

We quantified the expression of 51,900 genes using mRNA sequencing and identified 3363 proteins corresponding to 3360 unique genes through untargeted proteomics using six pooled samples (~50 larvae per pool). Thus, the transcriptomic and proteomic data were available for the same 3360 genes. To determine the primary sources of variation in mRNA and protein expression in larvae infected with entomopathogenic nematodes (EPNs), we conducted principal component analysis (PCA) on this shared gene set. Rank-normalization was applied to mitigate the impact of the outliers.

For mRNA expression, the first principal component (PC1) explained 81.81% of the variance (Figure 1A) and was strongly associated with infection status (*p* < 2.04 × 10^−11^), while the second principal component (PC2) accounted for only 4.87% of the variance and showed no significant association with infection (*p* = 0.97). Similarly, for protein expression, PC1 accounted for 45.25% of the variance (Figure 1B) and was significantly linked to infection (*p* = 0.0108), whereas PC2 explained 19.62% of the variance and was not significantly associated (*p* = 0.736). The PCA analysis of both the transcriptome and proteome indicates that PC2 does not have a significant association with infection status and is likely to reflect an individual-level biological variation or technical noise not captured by the known experimental variables. These findings indicate that EPN infection substantially alters both gene and protein expression patterns, as evident from the clear separation between infected and control groups along PC1 in both datasets.

### 3.2. Transcriptomic Analysis

Differential expression analysis using DESeq2 identified 1091 significantly altered transcripts following *S. carpocapsae* infection (log_2_ fold change > 1, adjusted *p* < 0.05) (Appendix A), with 893 transcripts downregulated and 198 upregulated. The volcano plot clearly illustrates the distribution of these differentially expressed genes (DEGs), highlighting both significantly up- and downregulated genes (Figure 2).

Gene Ontology (GO) enrichment analysis using ClusterProfiler revealed distinct patterns in gene expression associated with biological processes (BP), cellular components (CC), and molecular functions (MF). Among the 893 downregulated transcripts, the significantly enriched GO terms included “translation”, “cell redox homeostasis”, “carbohydrate metabolic process (BP)”; “integral component of membrane”, “cytoplasm”, and “mitochondrion (CC)”; and “ATP binding”, “monooxygenase activity”, and “metal ion binding (MF)” (Figure 3). Further analysis indicated that among these downregulated transcripts, 90 were linked to immune responses, particularly those involved in reactive oxygen species (ROS) scavenging, oxidative stress regulation, and antimicrobial peptide and immune enzyme secretion. Additionally, 151 downregulated genes were associated with metabolic pathways, including glycolysis and gluconeogenesis, while 103 were related to signal transduction, specifically those with functions involving metal ion binding.

Upregulated transcripts were predominantly associated with stress and immune responses such as “protein folding”, “oxidoreductase activity”, and “unfolded protein binding” (Figure 3). Specifically, 25 were linked to immune responses, particularly chaperone-mediated stress responses and redox enzyme activity regulation. Furthermore, 151 upregulated genes participated in metabolic pathways, primarily in ATP-binding processes, while 22 were associated with signal transduction (metal ion binding).

Kyoto Encyclopedia of Genes and Genomes (KEGG) enrichment analysis further delineated the host’s immune response, identifying significant alterations in pathways such as “Toll and Imd signaling”, “Phenylalanine metabolism”, “Glycolysis/Gluconeogenesis”, and “Oxidative phosphorylation”. Notably, the Toll and Imd signaling pathway, crucial for insect innate immunity, exhibited the highest significance among immune-related pathways (Figure 4). Collectively, these data indicate a comprehensive reprogramming of host immune defense, metabolic processes, and signal transduction in response to *S. carpocapsae* infection.

To further examine the infection-induced changes in gene expression, we performed an overrepresentation analysis (FDR < 0.05, *p* < 0.05) and identified 863 significantly enriched genes. The top 100 overexpressed genes were involved in key biological processes such as the immune response and metabolic regulation. Notably, genes associated with antimicrobial peptide production (e.g., *AMP1*, *Defensin*) and stress response (*HSP70*, *Catalase*) exhibited upregulation, reflecting an active host defense response to *S. carpocapsae* infection (Figure 5).

To validate the reliability of our transcriptomic data, we performed qRT-PCR on 16 differentially expressed genes using biological replicates. The qRT-PCR results were consistent with the mRNA-seq data, supporting the robustness of our findings (Figure 6).

### 3.3. Proteomic Analysis

Protein abundance data were analyzed using a linear mixed-model ANOVA, with rank-normalization applied prior to statistical testing. Proteins exhibiting >1.5-fold upregulation or <1/1.5-fold downregulation (adjusted *p* < 0.05) were classified as differentially expressed (Appendix A). In total, 150 significantly altered proteins were identified following *S. carpocapsae* infection, with 105 upregulated and 45 downregulated (Figure 7).

Among the most significantly upregulated proteins, *NOTCH3* (*p* = 0.0175) and *Sftpd* (*p* = 0.0191) were associated with immune response, tissue repair, and extracellular signaling. Conversely, the most significantly downregulated proteins, including *PXN* (*p* = 0.0127) and *CELA2A* (*p* = 0.0469), were linked to cell adhesion, migration, and metabolic regulation, suggesting a decline in cellular activity and metabolic demand during infection (Figure 8).

To explore the protein interaction networks, we analyzed differentially expressed proteins using the STRING database (v10.5) with a confidence threshold of >0.7. A network constructed from the top 50 interacting proteins revealed that ribosomal proteins (e.g., *RpL5*, *RpS8*, *Rpl27*) were upregulated, whereas cytochrome c proteins (e.g., *Cyt-c*) were downregulated. Functional enrichment analysis categorized differentially expressed proteins into three major modules: (i) protein synthesis (*Rpl5*, *RpL19*), (ii) stress response (*Hsp70Ba*, *Cat*, *ALDH4A1*), and (iii) metabolic regulation (*Tktl2*, *Cyc*). GO enrichment analysis identified 50 significantly enriched terms (adjusted *p* < 0.05), showing an increased expression of the proteins involved in extracellular matrix organization, tissue structure maintenance, embryonic development, germ cell differentiation, stem cell function, and ribosomal activity. In contrast, proteins related to adenylosuccinate synthase and protein phosphatase regulation were downregulated (Figure 9).

KEGG pathway enrichment analysis revealed significant enrichment in several biological pathways related to the host responses, notably in ECM–receptor interaction, protein digestion and absorption, and Epstein–Barr virus infection pathways. Among these pathways, the ECM–receptor interaction pathway exhibited the highest fold enrichment, suggesting that interactions between the extracellular matrix and cellular receptors play a critical role during host infection responses. Additionally, significant enrichment was observed in the metabolic and signaling pathways, such as glycosaminoglycan degradation, insulin secretion, VEGF signaling, and apoptosis pathways, highlighting their roles in the regulation of immune response and cell survival during nematode infection (Figure 10). These findings collectively underscore the complexity of the host’s proteomic responses to EPN infection, involving extensive remodeling of extracellular interactions, metabolic processes, and cell-signaling networks.

### 3.4. Comparison of mRNA and Protein Expression

To assess the correlation between the mRNA and protein expression changes following EPN infection, we applied a false discovery rate (FDR < 0.1) and identified 3363 proteins with corresponding mRNA data. Among these, 150 proteins were differentially expressed (upregulated: LFQ intensity ratio > 1.5, adjusted *p* < 0.05; downregulated: LFQ intensity ratio < 1/1.5, adjusted *p* < 0.05). Correlation analysis, based on a fold-change conversion to z-scores, revealed a significant but weak positive correlation (r = 0.062, *p* < 2.30 × 10^−2^).

Genes were categorized based on expression consistency: 39 genes showed concordant upregulation, 26 displayed consistent downregulation, while 10 genes exhibited decreased mRNA expression but increased protein levels, and 36 showed increased mRNA but decreased protein levels. While most genes (~1.93%) displayed consistent changes, a substantial proportion (~1.37%) exhibited discordant mRNA-protein trends (Figure 11).

KEGG pathway enrichment analysis revealed that genes with consistently upregulated expression were involved in the pentose phosphate pathways, hormone synthesis, ribosome biogenesis, and protein processing in the endoplasmic reticulum. Conversely, consistently downregulated genes were linked to amino acid metabolism (*ALDH4A1*, *GPT*), carbohydrate metabolism (*PyK*, *GALM*), immune defense (*Cyt-c-p*), signal transduction (*CanB2*), and stress response (*Cyt-c-p*). Genes with decreased mRNA but increased protein levels were enriched in pathways related to energy metabolism, oxidative stress, and nervous system responses (*IST1*, *Hsp70Ba*, *SNX2*, *NDUFS7*, *Cyc1*, *Lrp2*), whereas genes with increased mRNA but decreased protein levels were primarily associated with tryptophan metabolism (Figure 12).

Mediation analysis of 3360 genes demonstrated that changes in mRNA expression did not significantly influence the protein expression patterns (Figure 13).

## 4. Discussion

In this study, we employed integrated proteomic and transcriptomic analyses to explore the molecular responses of *C. dieckmanni* larvae to infection by *S. carpocapsae*. Our findings revealed that EPN infection causes significant changes in gene and protein expression, which was evident from the distinct separation between the infected and control groups in the principal component analysis (PCA) of both the transcriptomic and proteomic datasets (Figure 1). The notable alterations in genes and proteins were associated with immune defense, cellular stress, and metabolism, which highlighted the coordinated response of *C. dieckmanni* to EPN invasion. Additionally, the integrated analysis indicated a correlation between mRNA and protein expression levels. However, many genes displayed inconsistent regulation, which suggests that post-transcriptional and post-translational regulatory mechanisms are crucial in shaping the host response to *S. carpocapsae*.

### 4.1. Immune Response of C. dieckmanni Larvae to S. carpocapsae Infection

Transcriptomic analysis revealed a significant upregulation of antimicrobial peptides (AMPs) in *C. dieckmanni* larvae following infection by *S. carpocapsae*, such as *defensins* and an Attacin-like gene, as well as stress-related genes (Figure 5). These genes encode frontline effector molecules of insect humoral immunity, potentially controlling bacterial proliferation after symbiotic bacteria are released into the host [28]. AMP induction serves as a hallmark of systemic immune activation in other insect-EPN systems. For instance, upregulation of AMP via Toll/Imd pathways in *Drosophila* larvae infected with *X. nematophila*-harboring *S. carpocapsae* was delayed until bacteria released at ~14 h post-infection [13]. Similarly, *S. carpocapsae* infection triggers significant transcriptional increases of Attacin-C1 and defensin-2B in *Octodonta nipae* [26,29]. These parallels suggest that AMP induction in *C. dieckmanni* represents an evolutionarily conserved response, likely triggered by microbial pattern recognition (e.g., *X. nematophila*-derived peptidoglycans) following hemocoel invasion by the nematode–bacteria complex. AMP expression regulated by the NF-κB signaling pathways (Toll/Imd) activates during pathogen infection [30]. However, in our results, there is no significant upregulation of the Toll/Imd pathway components at the protein level based on proteomic data, which indicates that either immune evasion by *S. carpocapsae* or transient pathway activation is undetected. Whereas certain EPNs (such as *S. hermaphroditum*) actively suppress host immunity by inhibiting the Toll/Imd target genes in *Drosophila* via immunosuppressive effectors from *X*. *griffiniae* [26,31]. Therefore, the interaction between *S. carpocapsae* and its host appears multifaceted, although AMP responses are elicited upon infection, while certain secreted factors may concurrently suppress specific immune components [32,33]. For example, AMP expression was induced, and phenoloxidase activity was suppressed in *Drosophila* by *S. carpocapsae* secretions, but *X. nematophila* still inhibits phagocytosis and melanization [32,34]. The upregulation of defensin/Attacin without concomitant Toll/Imd protein elevation may reflect such immunomodulation.

Potential explanations include: (1) delayed immune recognition until bacterial release (minimal early NF-κB activation), (2) selective blockade of Toll/Imd signaling, or (3) technical limitations in detecting low-abundance immune proteins (e.g., NF-κB transcription factors or small secreted AMPs) [35,36]. Nevertheless, the marked AMP transcript accumulation indicates pronounced humoral immunity in infected *C. dieckmanni* larvae, which likely inhibits bacterial proliferation in the host [37].

The stress-related chaperones and antioxidant enzymes, particularly HSP70 and catalase, upregulate in infected *C. dieckmanni* larvae. An HSP70 elevation indicates that cytoprotective chaperone responses were triggered by pathogen-inducted proteotoxic stress, which is a common feature of insect–microbe interactions [38]. Zhang et al. (2025) proposed that microbial infection activates stress pathways (e.g., HIF-1α) in immunocytes, driving metabolic reprogramming and HSP expression [39]. HSPs expected for housekeeping functions may enhance immune signaling by stabilizing transcription factors, such as HSP70, promoting AMP production [40]. The coordinated induction of *defensins*, Attacin, HSP70, and antioxidant genes in *C. dieckmanni* exhibits an integrated defense strategy. Our results suggest that direct antimicrobial effectors are deployed alongside cellular protection mechanisms with an upregulated transcripts cluster within immune effector and stress response categories to maintain homeostasis during infection in *C. dieckmanni* [41,42].

### 4.2. Metabolic and Cellular Changes During Infection

A prominent trend in the transcriptome of nematode-infected weevil larvae was the downregulation of metabolic genes, particularly those involved in carbohydrate catabolism and energy production. Significantly suppressed transcripts were enriched in pathways including glycolysis, tricarboxylic acid (TCA) cycle, and oxidative phosphorylation, suggesting systemic metabolic suppression or redirection induced by *S. carpocapsae* infection. This aligns with a resource reallocation strategy, i.e., the “immune privilege” model where infected insects prioritize immune defense over growth or energy storage; for example, infection triggers metabolic shifts with channel nutrients toward immune molecule synthesis in fat bodies and immunocytes by depriving non-immune tissues (e.g., muscles, gut) [43]. Our results support the “immune privilege” model that *C. dieckmanni* larvae likely reduce energy expenditure in basal metabolism by downregulating enzymes in core metabolic pathways to support costly immune responses. Similar metabolic repurposing has been observed in *Drosophila* and other insects, where fat bodies switch from anabolic functions (e.g., lipogenesis) to immune secretion and suppress the TCA cycle gene to generate immune effectors during pathogen infection [44,45].

Warburg effect-like metabolic reprogramming was also identified. Aerobic glycolysis (Warburg metabolism) is upregulated to meet rapid ATP and biosynthetic precursor demands in pathogen-activated mammalian macrophages despite a lower ATP yield per glucose molecule [39,45]. Analogous strategies in insect immunity are mediated by activating immunocytes, which enhance glycolytic flux and pentose phosphate pathway activity while suppressing mitochondrial respiration [43,46]. The upregulated metabolic regulators, including *Tktl2* (transketolase) and *Cyc* (a putative cell cycle/metabolic protein), in infected *C. dieckmanni* larvae support this paradigm in our research. Transketolase is a key enzyme in the pentose phosphate pathway, providing NADPH for maintaining redox balance and ribose-5-phosphate for nucleotide synthesis [47]. Increased abundance of transketolase suggests metabolic rerouting to support immune protein production and oxidative stress management [48]. Concurrent downregulation of TCA cycle enzymes (observed at transcriptional levels) indicates reduced oxidative phosphorylation flux due to a host’s metabolic shift toward glycolysis. These findings align with previous research in various insect–pathogen systems, highlighting conserved immunometabolic strategies employed by insects in response to pathogen invasion. Notably, although this metabolic reprogramming enhances resistance, prolonged metabolic trade-offs may compromise host growth and fitness, reflecting a survival–growth balance during acute infection [43].

Furthermore, the genes associated with cellular structure and repair are downregulated after infection. Transcriptomic and proteomic analyses revealed significant suppression of cell adhesion and cytoskeletal regulators, including paxillin (PXN), cell adhesion molecule 2A (CELA2A), and Cdc42, which are critical for cell migration, wound healing, and gut integrity [49,50,51]. Reduced expression suggests compromised tissue repair processes, potentially resulting from either pathogen-induced damage surpassing the repair capacity of the host or a strategic reallocation of resources that prioritizes immune defense over tissue maintenance. Previous studies have documented similar pathogen-mediated disruptions in midgut regeneration and cuticle repair in insects, particularly when physiological resources are redirected toward immune responses [52,53]. Although *S. carpocapsae* infection induces internal injuries through nematode tissue penetration and bacterial lytic enzymes, pathogen-derived factors may also actively suppress host-healing mechanisms. For instance, certain EPN-associated bacteria secrete anticoagulants or antimelanization factors, thereby interfering with the wound closure processes of the host [54]. Therefore, the observed suppression of adhesion and tissue repair genes in *C. dieckmanni* likely exerts dual effects: first, enabling the host to prioritize resource allocation toward immune defense, and second, reflecting pathogen-driven interference with wound-healing processes. This strategy maintains a compromised internal environment, facilitating the proliferation of both nematodes and bacteria [11,55].

### 4.3. Proteomic Insights into Host Responses

The proteomic analysis provided complementary insights into the molecular mechanisms underlying the interaction between weevils and *S. carpocapsae*. The results demonstrated robust engagement of the host immune system at the protein level with concurrent metabolic alterations. The most upregulated proteins in infected larvae were *NOTCH3* and surfactant protein D (*Sftpd*). Notch signaling is associated with hemocyte differentiation and antimicrobial defense in insects and plays critical roles in immune regulation and developmental processes [56]. The elevation of *NOTCH3* in *C. dieckmanni* suggests activation of Notch-mediated signaling cascades as part of its anti-nematode response, potentially influencing hemocyte proliferation or immune gene expression.

*Sftpd*, which binds and clears pathogens in vertebrates, has no confirmed homologs in insects [57]. However, our study identified the upregulation of an *Sftpd*-like protein, raising the intriguing hypothesis that insects may utilize collectin-like molecules for pathogen recognition. This putative *Sftpd* homolog may bind bacterial surface components (e.g., *Xenorhabdus* cell envelope molecules), promoting their aggregation or facilitating phagocytic recognition, function analogous to the immunoregulatory roles of vertebrate *Sftpd* in pulmonary immunity [58,59]. While speculative, this finding opens new avenues for functional investigations into insect *Sftpd*-like proteins. Collectively, the proteomic data highlight the involvement of non-canonical immune components (e.g., *Notch*, *Sftpd*) and classical Toll/Imd pathways in coordinating responses to EPN infection. However, the specific roles of these non-traditional immune factors in insects remain to be validated.

Another notable proteomic trend was the altered expression of ribosomal and translational machinery components. Multiple ribosomal proteins (e.g., RpL5, RpS8, Rpl27) were coordinately upregulated in infected larvae, suggesting an enhanced capacity for protein synthesis. This likely reflects the increased demand for immune effectors and acute-phase proteins during infection [60]. Similar ribosomal upregulation has been observed in pesticide-resistant mosquitoes, where enhanced ribosome biogenesis supports the maintenance of cellular functions under toxic stress [61]. In our system, elevated ribosomal proteins may also contribute to tissue repair by replacing damaged proteins or represent an innate immune strategy, as certain ribosomal proteins are known to regulate immune signaling pathways. Recent studies have revealed dual roles for insect ribosomal proteins in mediating both stress responses and immune regulation [62]. The enrichment of ribosomal proteins in differentially expressed networks suggests both increased translational activity and potential ribosome-mediated immunomodulation. Notably, ribosomal protein induction is a conserved response across insects exposed to pathogens or pesticides, highlighting their roles in cellular maintenance and defense [61,63].

Conversely, cytochrome c (*Cyt-c*), a key component of the mitochondrial electron transport chain and a regulator of apoptosis, was downregulated in certain infected larvae [64]. While the release of *Cyt-c* limits pathogen replication by promoting apoptosis of infected cells in other insect–microbe systems [65]. For instance, *Bacillus cereus* and *Listeria monocytogenes* infections trigger *Cyt-c*-mediated apoptosis in insects [46,66]. However, *S. carpocapsae* infection-induced metabolic adaptations (e.g., suppressed glycolysis/TCA cycle enzymes) through the upregulation of regulatory proteins (*Tktl2*, *Cyc*), which suggests energy conservation serves as a survival strategy [43]. EPNs likely preserve host tissue integrity to support nematode development and bacterial proliferation by suppressing *Cyt-c*-mediated apoptosis. This is consistent with the fact that *X. nematophila* suppresses hemocyte responses and inhibits apoptosis by secreting immunomodulators, such as phospholipase A2 inhibitors [55,67]. Similar anti-apoptotic strategies are also employed by parasitoid wasps and viruses to maintain host viability [68,69,70,71]. Our proteomic findings collectively suggest a profound physiological shift in infected hosts, characterized by the upregulation of immune-related proteins, including understudied candidates like *Notch* and *Sftp*, amplification of translational machinery, reconfiguration of metabolic enzymes, and suppression of pro-apoptotic factors. When considered alongside transcriptional trends, these results illustrate how *C. dieckmanni* balances robust immune defense with the maintenance of cellular homeostasis during EPN infection.

### 4.4. Discrepancies Between mRNA and Protein Expression

Comparative analysis of transcriptomic and proteomic datasets revealed frequent discordance between mRNA and protein levels, highlighting the complexity of regulatory processes during infection. Although the overall directional trends of major functional categories, such as the suppression of metabolic pathways and upregulation of immune effectors, are consistent between mRNA-seq and proteomic data, which supports the reliability of our multi-omics approach, many individual genes showed divergent regulation. For example, certain transcripts demonstrated significant induction without corresponding alterations at the protein level, whereas others displayed upregulated proteins despite no change in mRNA expression. A subset even exhibited opposing trends between transcript and protein levels. Such mRNA-protein mismatches have been widely reported in insect immunity studies, reflecting the influence of post-transcriptional mechanisms (e.g., mRNA stability and translational efficiency) and post-translational factors (protein localization, modification, and degradation) on protein abundance [72]. These findings underscore that mRNA expression alone is not a reliable predictor of functional protein levels in immune-activated cells.

For instance, multiple AMP genes (e.g., *defensins* and *attacins*) are upregulated at the transcriptional level, whereas their corresponding peptides do not exhibit significant changes in protein abundance. Such discrepancies could reflect the rapid secretion of these small peptides into the hemolymph or indicate post-transcriptional regulation, such as delayed or suppressed translation [73]. Similarly, transcripts encoding certain signaling molecules were induced, though their protein products may require post-translational activation or exhibit only transient presence [74]. A notable example from our data is the NF-κB inhibitor Cactus; its mRNA displays moderate downregulation (suggesting pathway activation), and no significant change in Cactus protein levels is observed. This could be attributed to rapid protein degradation followed by resynthesis.

Pathogen-induced regulatory strategies may also contribute to the observed mRNA-protein mismatches. Pathogens may allow host mRNA synthesis (potentially as decoys) while subsequently inhibiting translation or accelerating protein degradation [75]. In the *S. carpocapsae*–weevil interaction, when hosts upregulate certain immune transcripts, nematode symbionts may suppress these immune responses by producing proteases or other factors that target immune proteins [8]. A similar observation was made in *X. nematophila*-infected Spodoptera frugiperda larvae, where bacterial interference suppressed the accumulation of cecropin AMP accumulation despite transcriptional induction [67]. This highlights that mRNA quantification alone may overestimate actual host defense response.

The specificity of tissue or cell type further complicates the interpretation of immune responses. Whole-larva homogenates used for mRNA and proteomic analyses likely blended signals from multiple tissues. Immune gene transcripts highly upregulated in fat body cells might encode proteins that are predominantly active in hemocytes or vice versa [76]. Such spatial segregation could lead to apparent mismatches when interpreting whole-organism averages. For example, certain immune regulators expressed in fat bodies of *Drosophila* during EPN infection become functionally active only upon secretion or subsequent uptake by hemocytes [43]. Temporal or spatial mismatches between mRNA and protein detection may thus generate discrepancies.

Comparative pathway analysis revealed general concordance between upregulated genes and proteins in carbohydrate metabolism, protein synthesis, and extracellular matrix–receptor interaction pathways. However, more genes exhibited divergent mRNA-protein trends, with some showing inverse regulation [77,78]. Although it is often assumed that mRNA levels directly reflect protein function in the transcriptomic studies of EPN–host interactions, our data reveal frequent discrepancies between transcript and protein abundance. The molecular mechanisms underlying host resistance to *S. carpocapsae* infection are more complex than direct transcriptional regulation. Protein expression changes following infection are not solely mediated by mRNA levels but are also shaped by cell-type-specific factors, including post-transcriptional modifications, protein stability, translational efficiency, and post-translational regulation [79,80].

Therefore, it is necessary to integrate multiple approaches to comprehensively understand the host’s immune response to nematode infection, as highlighted by the transcriptome–proteome discrepancies observed in our study. Relying solely on mRNA profiles may be inadequate, as the robust gene-level responses at the transcript level may not always be reflected in protein expression, and conversely, changes in protein levels may not correlate with mRNA abundance. Multi-omics integration is critical in EPN studies, given the substantial impact of post-transcriptional regulation on transcriptome-based predictions [38].

The immune response of *C. dieckmanni* to *S. carpocapsae* results from a combination of transcriptional changes and subsequent regulatory modifications. In response to nematode infection, hosts trigger conserved antimicrobial pathways (Toll/Imd) and immunometabolic shifts, including production of AMP, induction of chaperone, and reallocation of metabolic resources [13,39,43]. Understanding discordance between mRNA and protein expression is essential for elucidating the insect resistance response to EPNs and optimizing their biocontrol applications [81]. While we focused exclusively on short-term responses to *S. carpocapsae* infection, long-term infections may trigger different physiological adaptations. Future investigations should explore long-term host adjustments to better understand the defensive mechanisms.

## 5. Conclusions

Our integrated transcriptomic and proteomic analyses revealed extensive molecular alterations in *C. dieckmanni* larvae following infection by *S. carpocapsae*. Host immune responses were characterized by significant upregulation of immune-related genes and proteins, including antimicrobial peptides (AMPs) and stress-responsive proteins, reflecting active defense mechanisms. Additionally, infection induced substantial metabolic reprogramming, including the suppression of carbohydrate metabolism and strategic reallocation of metabolic resources. Notably, discrepancies between mRNA and protein expression suggested complex regulatory mechanisms involving post-transcriptional and post-translational processes. In summary, these findings enhance our understanding of insect responses to entomopathogenic nematodes and provide valuable insights for optimizing biocontrol strategies. Nevertheless, further research addressing the long-term physiological adaptations and species-specific responses is necessary to generalize and fully utilize these findings.

## Figures and Tables

**Figure 1 insects-16-00503-f001:**
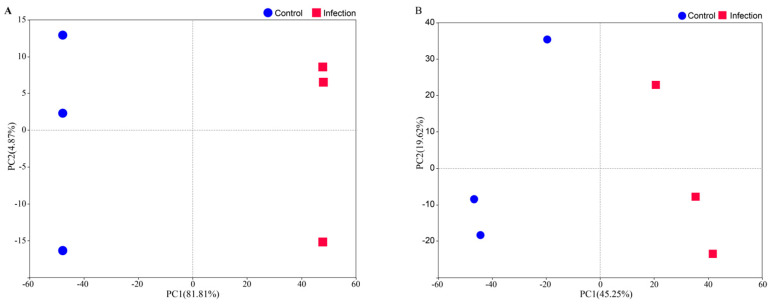
PCA of mRNA-seq (**A**) and proteomics (**B**) data. Each point represents a biological replication.

**Figure 2 insects-16-00503-f002:**
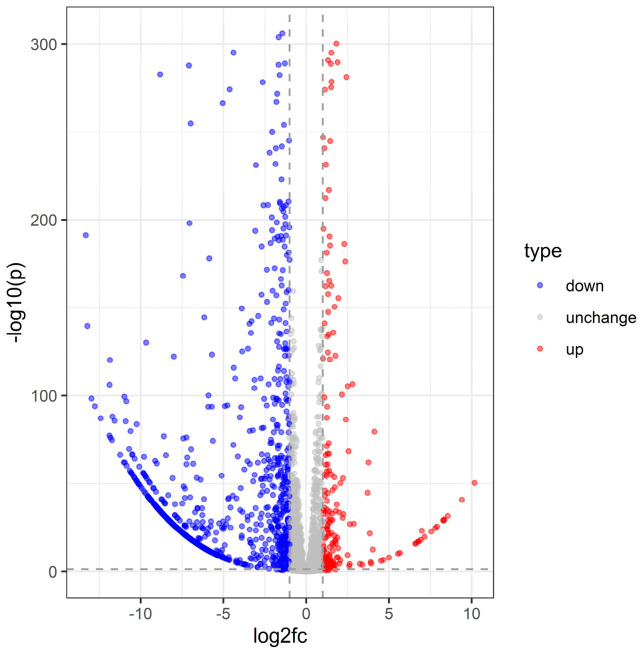
Volcano plot of *C. dieckmanni* DEGs infected with *S. carpocapsae*.

**Figure 3 insects-16-00503-f003:**
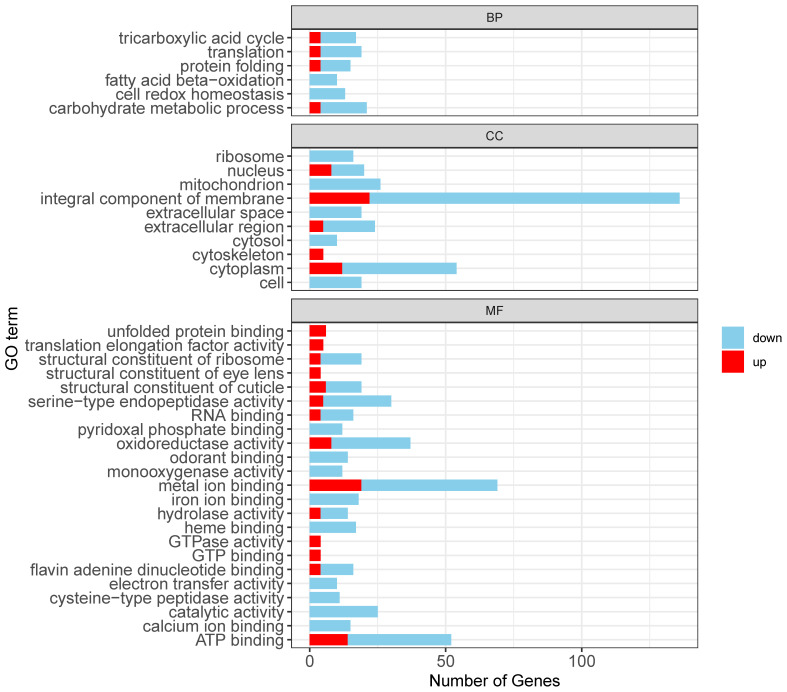
Functional enrichment analysis of differentially expressed transcripts.

**Figure 4 insects-16-00503-f004:**
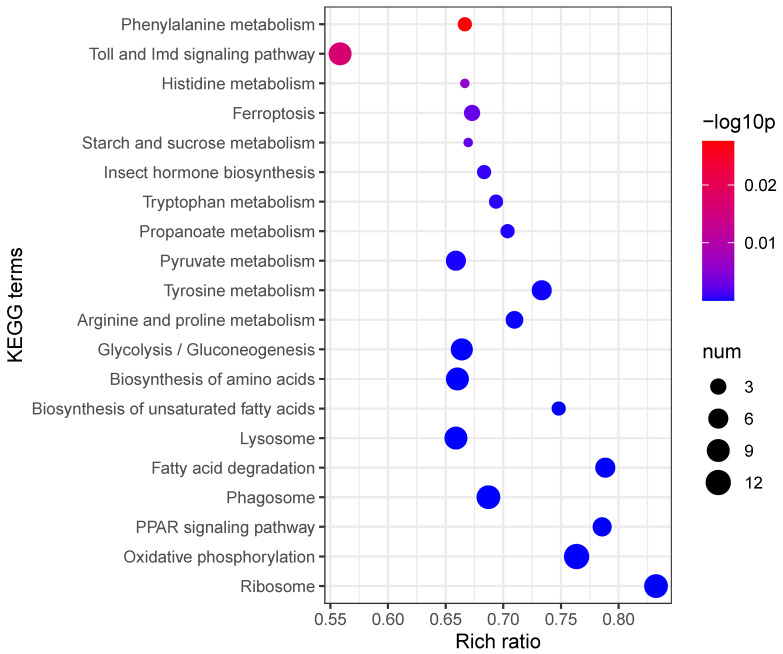
KEGG enrichment analysis of differentially expressed transcripts.

**Figure 5 insects-16-00503-f005:**
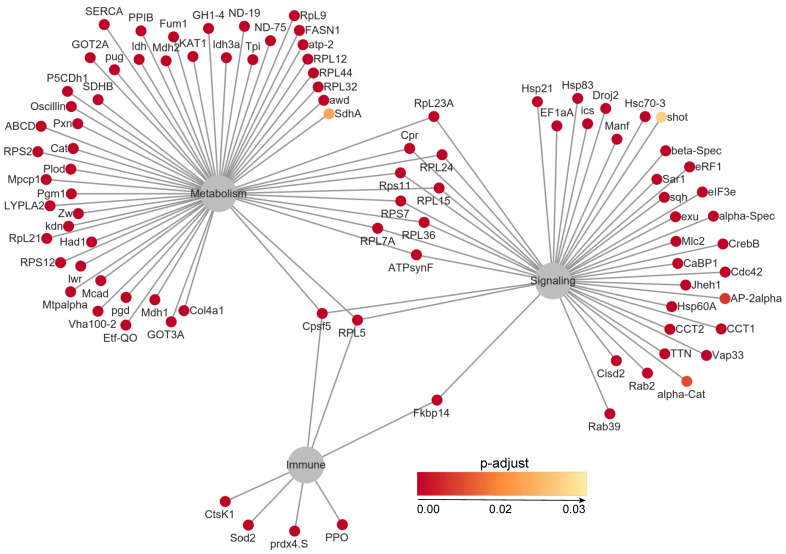
Network analysis of the top 100 overrepresented genes.

**Figure 6 insects-16-00503-f006:**
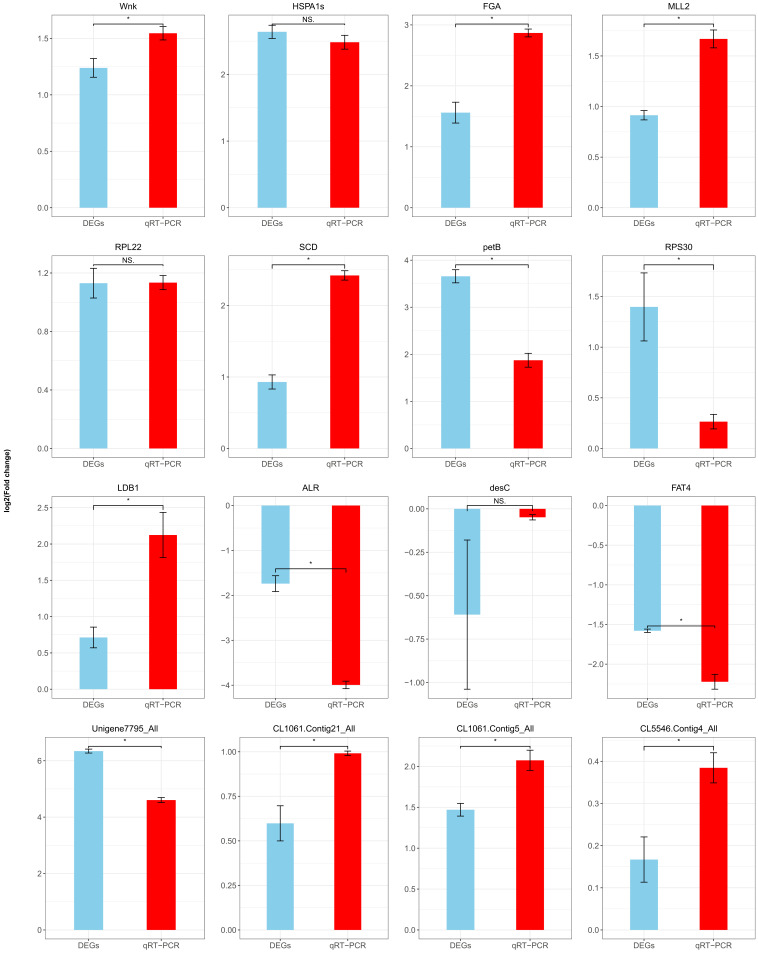
qRT-PCR validation of differentially expressed genes (DEGs). The difference between DEGs and the genes of qRT-PCR was assessed using a Student’s *t*-test (*p* < 0.05). Data are shown as means ± SEM. * indicates a significantly different at the level of *p* < 0.05, and NS is not significantly different (*p* < 0.05).

**Figure 7 insects-16-00503-f007:**
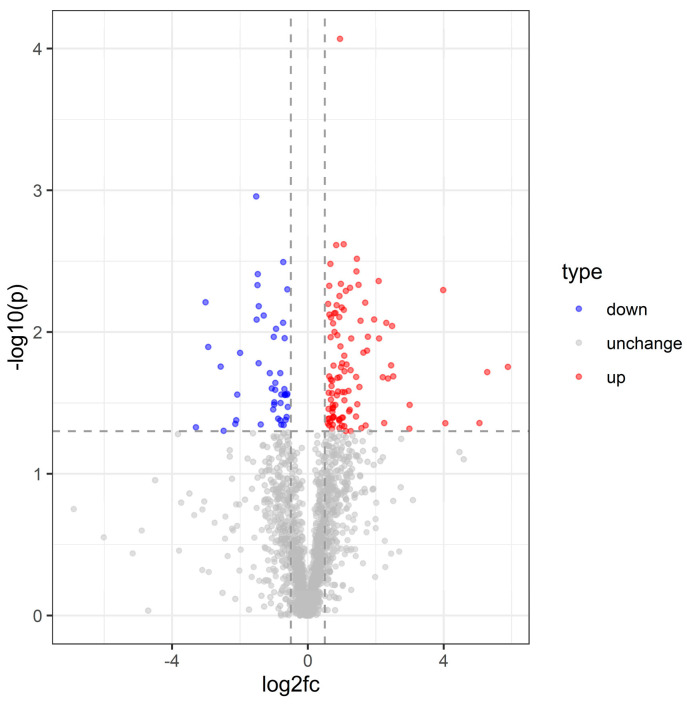
Volcano plot of *C. dieckmanni* DEPs infected with *S. carpocapsae*.

**Figure 8 insects-16-00503-f008:**
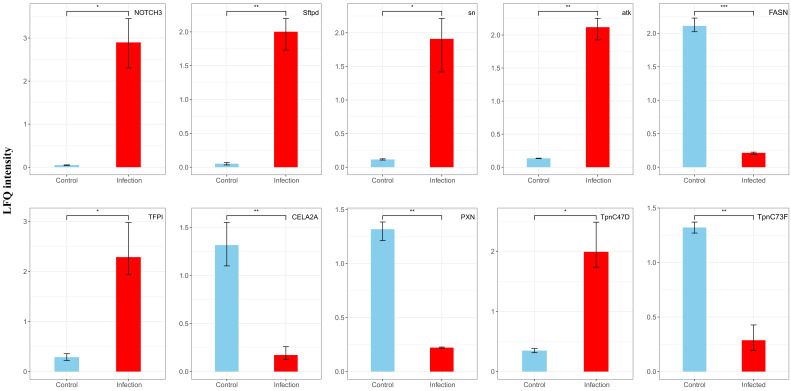
Changes in LFQ intensity (ratio) of the top 10 differentially expressed proteins before and after EPN infection. The DEPs was assessed using a Student’s *t*-test (*p* < 0.05). Data are shown as means ± SEM. * indicates a significantly different at the level of *p* < 0.05, ** indicates a significantly different at the level of *p* < 0.01, *** indicates a significantly different at the level of *p* < 0.001.

**Figure 9 insects-16-00503-f009:**
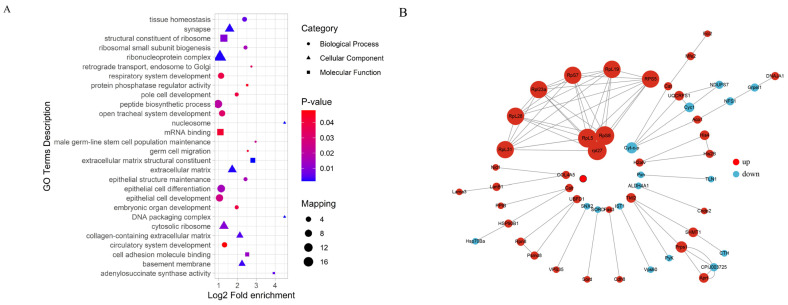
Functional enrichment analysis of differentially expressed proteins (**A**) and network analysis (**B**).

**Figure 10 insects-16-00503-f010:**
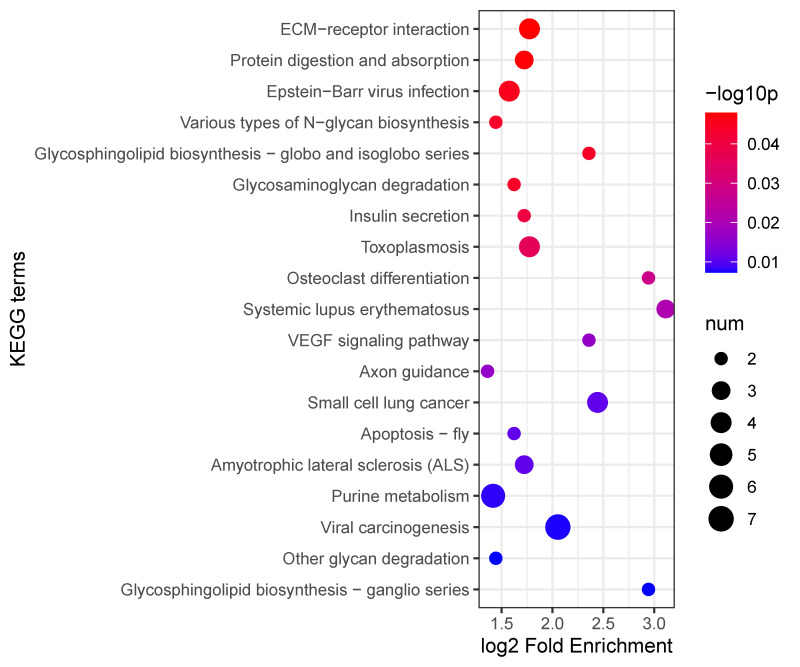
KEGG enrichment analysis of differentially expressed proteins.

**Figure 11 insects-16-00503-f011:**
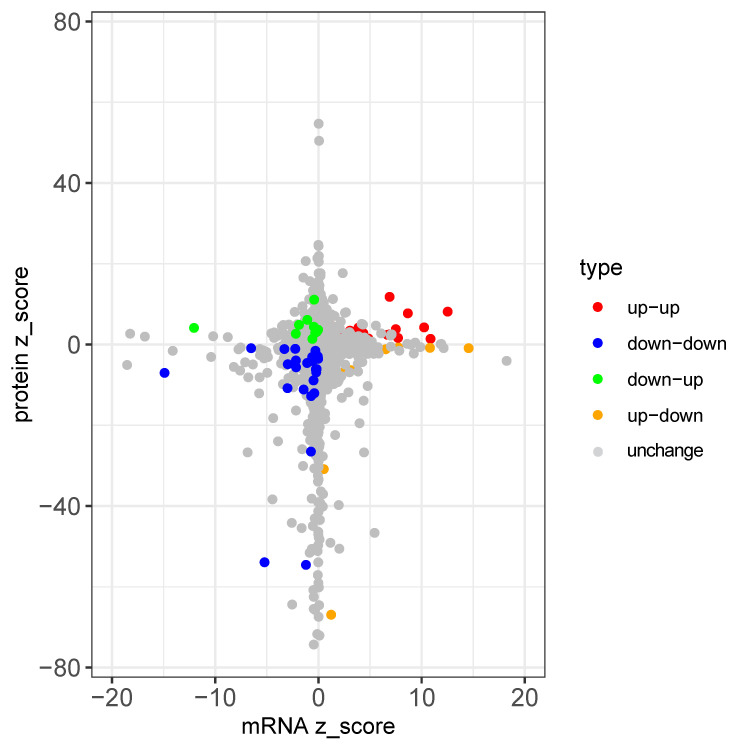
mRNA-protein expression comparison.

**Figure 12 insects-16-00503-f012:**
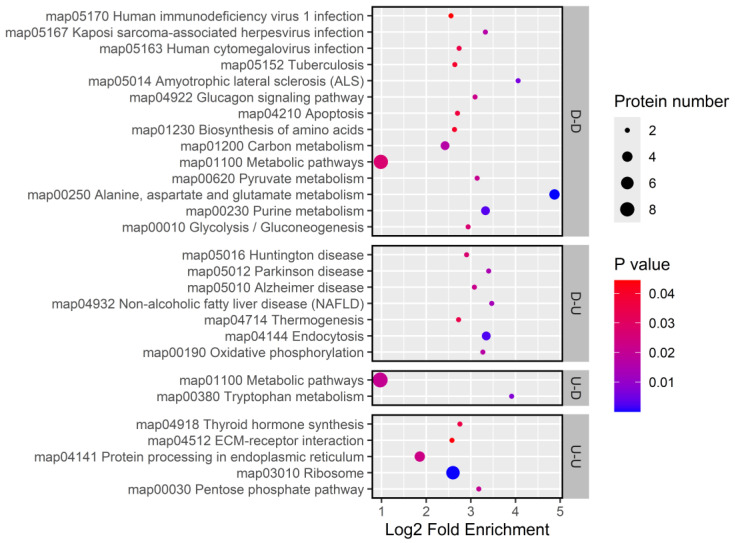
KEGG enrichment for genes with concordant and discordant mRNA-protein expression changes.

**Figure 13 insects-16-00503-f013:**
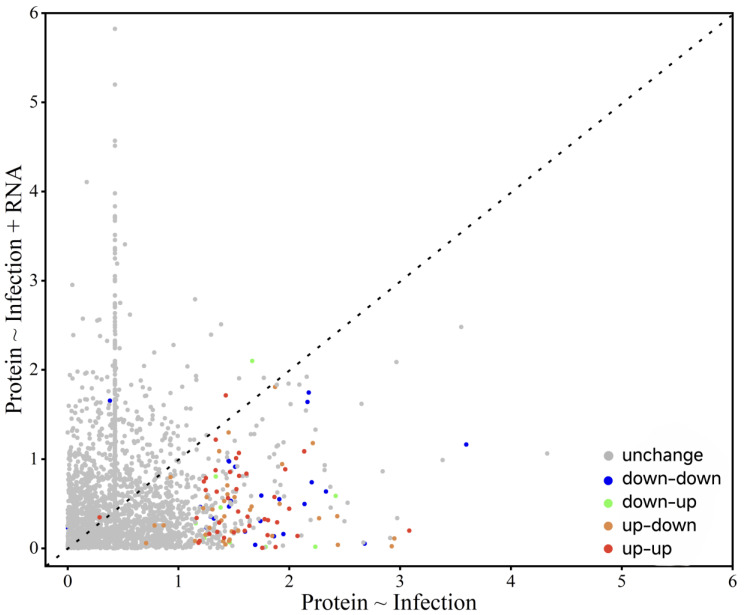
Mediation analysis of EPN infection-induced protein expression changes.

## Data Availability

The raw data supporting the conclusions of this article will be made available by the authors upon request.

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
