# Peer review of "Multi-Omics Analysis of Curculio dieckmanni (Coleoptera: Curculionidae) Larvae Reveals Host Responses to Steinernema carpocapsae Infection"

_insects, 2025, doi:10.3390/insects16050503_

Round 1
Reviewer 1 Report
Comments and Suggestions for Authors
This research employed transcriptomics and proteomics techniques to investigate the response of hazelnut weevil larvae (Curculio dieckmanni) upon infection by the entomopathogenic nematode Steinernema carpocapsae. The results revealed significant upregulation of immune-related transcripts (such as antimicrobial peptides and stress-respon sive proteins like heat shock protein 70) and an overall correlation between mRNA and protein expression levels. The quality of the thesis is moderately satisfactory.
The greatest query I have regarding this research lies in the source of the experimental insects: The author employed insects collected from the natural environment as the source of test insects. How can it be guaranteed that the source of test insects has not been infected by nematodes or other pathogens? The author employed insects collected from the natural environment as the source of test insects. How can it be guaranteed that the source of test insects has not been infected by nematodes or other pathogens?
Minor revision suggestions:
Line 3: Host Responses →immune responses
Line 25: biological control agent
Line 42-43: Entomopathogenic nematodes (EPNs) are widely used as biological control agents for soil-dwelling insect pests.
Author Response
Dear Editor and Reviewer,
Thank you for your thorough review and insightful comments. We have carefully considered each suggestion and revised the manuscript. We address each comment individually and detail the corresponding changes made to the manuscript:
- Comment: The author employed insects collected from the natural environment as the source of test insects. How can it be guaranteed that the source of test insects has not been infected by nematodes or other pathogens?
Response: We sincerely appreciate the reviewer’s concerns regarding potential limitations in the selection of experimental materials and design. The host insect, Curculio dieckmanni larvae, cannot currently be mass-reared under laboratory conditions for successive generations. Therefore, we have to collect the larvae from the soil of the hazelnut orchard. To avoid being infected by natural nematodes or other pathogens, the larvae were collected during the diapause state and then stored in a constant-temperature incubator for approximately 15 days under natural temperature and humidity until the experiments began.
Usually, the host insects infected by entomopathogenic nematode (EPN) exhibit over 90% mortality within 48 hours and complete mortality within 72 hours (Gaugler, R. Entomopathogenic Nematology. 2002, Springer; Batalla-Carrera et al., 2010, BioControl 55: 523-530; Van Damme et al., 2016, Pest Manag. Sci. 72(9): 1702-1709). In addition, the hosts infected by pathogens generally perish within 7 days at most (Islam et al., 2023, Sci. Rep. 13(1): 8331).
The 15 days of incubation guaranteed that the test insects were free of infection by natural nematodes or other pathogens. We also double-checked the survival status and external morphology of the test larvae before the experiment.
- Comment: Line 3: "Host Responses" should be corrected to "immune responses."
Response: We sincerely appreciate the reviewer’s attention to terminological precision. The phrase "Host Responses" has been revised to "immune responses" in the text. We also revised the other places for consistency.
- Comment: Line 25: "biological control alternative" should be changed to "biological control agent."
Response: Thank you for highlighting this inaccuracy. We have replaced "biological control alternative" with "biological control agent" in the revised manuscript.
- Comment: Lines 42-43: "Entomopathogenic nematodes (EPNs) are widely used as bioinsecticides for the biological control of soil-dwelling insect pests" should be revised to "Entomopathogenic nematodes (EPNs) are widely used as biological control agents for soil-dwelling insect pests"
Response: We fully agree with the reviewer’s suggestion to improve clarity and precision. The sentence has been revised as recommended, enhancing both accuracy and readability.
Once again, we sincerely thank you for your meticulous review and constructive feedback, which have significantly improved the quality of our manuscript. Should any further revisions be required, we will gladly address them promptly.
Best regards,
Bin Wang
Reviewer 2 Report
Comments and Suggestions for Authors
The manuscript addresses a relevant topic, combining transcriptomic and proteomic approaches to investigate the molecular response of Curculio dieckmanni larvae to infection by Steinernema carpocapsae, an important entomopathogenic nematode used in biological control. However, despite the potential relevance of the study, the manuscript suffers from serious conceptual, methodological, and structural weaknesses.
Introduction
The introduction should focus more specifically on Steinernema carpocapsae, as it is the central entomopathogenic nematode (EPN) used in this study. There is excessive reference to EPNs in plural, often mixing Heterorhabditis bacteriophora and S. carpocapsae without discriminating between them. This generalization weakens the scientific context, as these species differ in virulence factors, infection dynamics, and host interactions. Furthermore, recent literature addressing the virulence mechanisms, immune evasion strategies, and host responses specifically to S. carpocapsae are omitted. This limits the depth and accuracy of the introduction in reflecting the current state of the art.
Line 44
“EPNs invade the host and rapidly release symbiotic bacteria (Photorhabdus or Xenorhabdus), which evade recognition by host immune hemocytes with inhibiting cellular..."
- This sentence requires improvement for clarity and scientific precision
Lines 47–50
“Subsequently, both EPNs and their symbiotic bacteria secrete venom proteins and virulence factors...”
- This passage oversimplifies the diversity and specificity of virulence factors secreted by EPNs and their bacterial symbionts. It relies on outdated references (1997 and 2004), omitting significant progress made in the last decade. For instance, several key virulence effectors from S carpocapsae. Including more recent literature would better support the claims and align with current scientific understanding.
Materials and Methods
Main Concerns
Several essential methodological details are omitted or insufficiently described:
- No information is provided about the equipment or software used for RNA integrity evaluation or transcriptome assembly.
- The transcriptome analysis pipeline (e.g., mapping algorithm, normalization, statistical thresholds) is not detailed.
- It is unclear whether the raw RNA-seq data were submitted to a public database; if submitted, the accession number is missing or not clearly indicated.
- For the proteomics analysis, details are sparse: no mention of the LC-MS/MS system used, acquisition mode, database searched (e.g., NCBI, UniProt), or search engine/software (e.g., MaxQuant, Mascot).
- major experimental bias in the control conditions: larvae injected with sterile water are compared with larvae injected with nematodes diluted in PBS. This introduces an osmotic and chemical inconsistency between groups and could confound the results.
Line 100
“The Steinernema carpocapsae nematode population was obtained from…”
- The specific strain of S. carpocapsae used is not identified. This is critical, as different strains can vary significantly in their virulence and in the immune responses they elicit in insect hosts. Full strain identification (e.g., All, Breton, UK1) should be included.
Line 103
“The nematodes were stored in 100 mL of sterile water…”
- Clarification is needed: what type of water was used (e.g., autoclaved distilled water, Milli-Q)?, what was the concentration of nematodes (IJs) per mL in storage? This is important for reproducibility.
Line 108
“A nematode suspension was prepared by diluting S. carpocapsae in PBS buffer to achieve a concentration of 2 nematodes per milligram of weevil body weight.”
- This is an unusual metric. Typically, nematode dose is expressed as a fixed number of infective juveniles (IJs) per host, not per mg of host weight. This methodology should be clarified or justified with literature support.
Line 113
“The control group were treated with an equivalent volume of sterile water.”
- Injecting insects with water may cause osmotic stress. Since the experimental group was injected with nematodes suspended in PBS, the control group should also have received PBS to ensure comparability.
Line 113 (continued)
“Each group consisted of at least 50 larvae, with a total of 132 larvae included in the study.”
- The number of biological replicates is not stated. It should be clarified how many larvae were used per replicate, how many replicates were analyzed, and how samples were pooled for transcriptomic and proteomic analysis.
Line 134
“Subsequent steps, including protein concentration measurement...”
- The proteomic methodology is described superficially. The authors should provide: LC-MS/MS platform used, acquisition settings, search database and software version, protein/peptide identification thresholds. Without these details, the reproducibility and interpretation of the proteomic data are compromised.
Results
Throughout the results, the authors frequently refer to “EPN infection” without clearly indicating that only S. carpocapsae was used. This imprecision weakens the interpretation and attribution of host responses.
The transcriptomic results are underexplored:
- No volcano plots or heatmaps are shown to illustrate differential expression.
- The number of up- and downregulated genes is only briefly mentioned.
- Gene Ontology (GO) enrichment is presented without detail on which biological processes, molecular functions, or pathways were most affected.
- KEGG pathway analysis is not clearly presented, and there is no discussion of immune-related pathways (e.g., Toll, Imd, JAK-STAT) that are typically implicated in insect responses to pathogens.
ine 165
“PC1 explained 81.81% of the variance… while PC2…”
- It is unclear what PC2 represents in this context. Does it reflect intra-group variability, biological variable?
Line 181
“The qRT-PCR results were consistent with the mRNA-seq data…”
- The qRT-PCR results are shown before the transcriptomic differential expression results, which is unconventional. Typically, qRT-PCR is used to validate a subset of the transcriptomic findings and is presented afterwards. The structure of the results section should be reorganized to improve logical flow.
Discussion
Several key findings are reiterated, but not critically analyzed or linked back to the methods and figures. Given the superficial exploration of the data in the results, the discussion lacks a solid foundation and fails to integrate current knowledge from recent publications in the field.
The manuscript requires substantial editing to improve the quality of English. There are frequent grammatical errors, awkward sentence constructions, and issues with verb tense, article usage, and word choice throughout the text.
Author Response
Dear Editor and Reviewers,
Thank you for reviewing our manuscript and providing valuable feedback. We have carefully revised the paper in response to each of your comments. We address your concerns point-by-point and detail the corresponding modifications.
- Reviewer Comment: The introduction lacks sufficient description of S. carpocapsae, overly generalizes EPN species, and omits recent studies.
Response: We appreciate the reviewer’s suggestion regarding the introduction. We have expanded the description of Steinernema carpocapsae in the introduction section, adding the biological characteristics and relevant research of application examples in the biocontrol of S. carpocapsae to enhance timeliness and focus (cited as [1], [2], [3], [4], [12] [14], [16], [19], [21], [22]). Additionally, the recent results about insect immune responses to nematode infection were cited in the revision (e.g., [5], [6], [7], [8], [13], [15], [20]). These revisions ensure the introduction is more specific to S. carpocapsae and reflects advances from the past five years. We rewrote the section of introduction, there are more specific details for the revision, and this may require the reviewer to spend additional time reviewing the revised manuscript to obtain more detailed information on the changes made. We sincerely apologize for any inconvenience this may cause.
- Reviewer Comment: The descriptions of virulent mechanisms and immune evasion cite outdated literature.
Response: We have updated the background sections on S. carpocapsae virulence and immune evasion with recent references. For example:
Garriga et al. (2020) demonstrated that S. carpocapsae and its symbiont evade host immune recognition during early infection ([5]).
Roy et al. (2020) highlighted symbiont-mediated immunosuppression via inhibition of the arachidonic acid pathway ([7]).
Lima et al. (2022) identified S. carpocapsae venom proteins modulating host immunity ([6]).
These additions reflect current understanding of virulence factors (e.g., venom proteins) and immune evasion strategies.
- Reviewer Comment: The Methods section lacks RNA/protein analysis details, equipment/software specifications, and inconsistent control group injections.
Response: We added the detailed description of the method and analytical protocols, including RNA extraction kit details (e.g., TRIzol reagent (Tiangen Biotech, Beijing, China)), sequencing platform (Illumina), software (DESeq2 v1.38.0 for RNA; MaxQuant v1.6.15.0 for proteomics), and LC-MS parameters.
Control group clarification: Corrected an oversight in describing the control group. Both groups received PBS buffer, with controls lacking nematodes. This has been supplemented in the revised manuscript.
- Reviewer Comment: Missing strain and concentration details for S. carpocapsae.
Response: We added the strain source in the revision. All strains of S. carpocapsae were purchased from Hongrun Agricultural Technology Co., China, and stored at 14°C, the concentration was ~1,000 IJs/mL in autoclaved water.
- Reviewer Comment: Nematode dosing by host weight (mg) is unconventional; clarify or cite supporting literature.
Response: The individual weight of Curculio dieckmanni larvae varied from 0.037 g to 0.103 g, to avoid the individual difference, we adjusted infection doses according to weight. This approach aligns with studies addressing host size variability.
- Reviewer Comment: Unclear biological meaning of PC2 in PCA.
Response: We are so sorry for the confusing description. PC1 (81.81% transcriptome, 45.25% proteome variance) reflects infection status. PC2 (4.87% transcriptome, 19.62% proteome) represents residual biological variation or technical noise unrelated to infection. This interpretation is now explicitly stated in the Results.
- Reviewer Comment: Insufficient presentation of omics data (volcano plots, GO/KEGG details, immune pathway analysis).
Response: Thank you for the suggestion. We added volcano plots for differential genes/proteins, expanded GO/KEGG results (e.g., immune-related terms like Toll/IMD pathways), and detailed immune pathway analyses (e.g., antimicrobial peptide upregulation via Toll signaling) in the revision.
- Reviewer Comment: Disordered Results section (qRT-PCR precedes main findings).
Response: We restructured the section of results to present transcriptomic/proteomic analyses first, and then followed by qRT-PCR validation for clarity.
- Reviewer Comment: Discussion lacks depth and connection to existing studies.
Response: We rewrote the discussion. First, we expanded the biological significance of the upregulation of immune-related genes. For example, we observed a marked upregulation of several antimicrobial peptides and heat shock proteins, which is consistent with the findings of Sanda et al. (2018) (The reference number in the manuscript is [31]), which reported immune gene upregulation in weevil larvae following S. carpocapsae infection. We emphasized the critical role of these antimicrobial molecules in defending against nematode-bacteria complex infections and supported our interpretation with recent studies.
Second, we discussed the potential reasons for the downregulation of metabolic pathways. We highlighted that insects often reallocate energy resources during immune responses by downregulating basal metabolism to support immune function—a phenomenon referred to as the “immunity-metabolism trade-off” (Dolezal et al., 2019. The reference number in the manuscript is [45]). The downregulation of genes related to carbohydrate metabolism in our results aligns well with this model, which we emphasized and compared with relevant literature in the revised discussion.
In addition, we incorporated several recently published studies, such as Garriga et al. (2023) (The reference number in the manuscript is [13]), to compare our findings with nematode infection responses observed in other species. For instance, we noted that the immune pathways activated in C. dieckmanni larvae share commonalities with those reported in Drosophila and noctuid larvae, while also exhibiting species-specific differences. This comparative approach helped to contextualize our findings and demonstrate how they enhance the current understanding of insect-nematode interactions.
Finally, we summarized the significance of our findings and outlined potential directions for future research. The revised discussion is more closely integrated with prior studies and provides a deeper interpretation of the biological implications of our findings.
- Reviewer Comment: Language requires significant improvement.
Response: The manuscript has been thoroughly edited for language by colleagues proficient in academic English to improve clarity and professionalism.
All revisions addressing these comments have been incorporated into the manuscript and are tracked in the revised version.
Thank you again for your constructive feedback. We believe these revisions have substantially strengthened the manuscript.
Sincerely,
Bin Wang
On behalf of all authors
Round 2
Reviewer 1 Report
Comments and Suggestions for Authors
After revision, the quality of the manuscript has been significantly improved.
Regarding Figure 6:
-
Statistical Method Used: What statistical method was applied in this figure? The statistical method should be indicated in the figure caption.
-
Unequal Error Bars: Why are the positive and negative error bars of unequal length? This issue requires further revision.
Author Response
Dear Editor-in-Chief,
We greatly appreciate the opportunity to revise and resubmit our manuscript entitled "Multi-Omics Analysis of Curculio dieckmanni (Coleoptera: Curculionidae) Larvae Reveals Host Responses to Steinernema carpocapsae Infection" for consideration in Insects. We have carefully addressed the additional comments raised by the reviewer, clearly highlighting all modifications made in the revised manuscript.
Below, we provide a detailed, point-by-point response to the reviewer’s comments:
1 Reviewer comment: What statistical method was applied in Figure 6? The statistical method should be indicated in the figure caption.
Response: We thank the reviewer for pointing out this omission. We have now clearly indicated in the caption of Figure 6 that the statistical significance of differential gene expression between RNA-seq data and qRT-PCR results was assessed using a student’s t-test (p < 0.05). We have also revised the Materials and Methods section (section 2.6 Real-Time Quantitative PCR, lines 256–258) to include an additional reference for the qRT-PCR calculation method and explicitly noted that data were log-transformed [log(2^-ΔΔCt)] to achieve normality.
- Reviewer comment: Why are the positive and negative error bars of unequal length? This issue requires further revision.
Response: We sincerely apologize for this oversight. After careful review, we identified an error in our R script where the error bars mistakenly represented the mean ± (max/min values) rather than the mean ± Standard Error of the Mean (SEM). We have rectified this issue by correcting our R script and recalculating the values accordingly. Figure 6 has been updated with appropriate, symmetric error bars reflecting the mean ± SEM. This correction has been highlighted clearly in the manuscript.
In addition to these revisions, we confirm:
All references cited are relevant and appropriate for the manuscript's context.
No references recommended by the reviewers were deemed unnecessary or irrelevant.
All revisions made in the manuscript are highlighted for easy identification.
We trust these revisions fully address the remaining concerns. We appreciate your careful consideration of our manuscript and look forward to your response.
Sincerely,
Bin Wang
wb2651747251@outlook.com